# Prevalence and Related Risk Factors of Vitamin D Deficiency in Saudi Children with Epilepsy

**DOI:** 10.3390/children9111696

**Published:** 2022-11-05

**Authors:** Reem Al Khalifah, Muddathir H. Hamad, Abrar Hudairi, Lujain K. Al-Sulimani, Doua Al Homyani, Dimah Al Saqabi, Fahad A. Bashiri

**Affiliations:** 1Division of Pediatric Endocrinology, Department of Pediatrics, College of Medicine, King Saud University Medical City, King Saud University, Riyadh 11461, Saudi Arabia; 2Division of Pediatric Neurology, Department of Pediatrics, College of Medicine, King Saud University Medical City, King Saud University, Riyadh 11461, Saudi Arabia; 3College of Medicine Research Center, King Saud University, Riyadh 11461, Saudi Arabia

**Keywords:** epilepsy, children, vitamin D, prevalence, risk factors, ASMs

## Abstract

Background: Vitamin D has a role in the pathogenesis of many medical disorders, especially those of the central nervous system. It is essential in maintaining the bone health of children. However, patients with epilepsy are at high risk of developing vitamin D deficiency due to antiseizure medications (ASMs). Therefore, we aimed to assess the prevalence of vitamin D deficiency and related risk factors in children with epilepsy. Methods: This is the baseline report of a pragmatic, randomized, controlled, open-label trial that assessed the impact of vitamin D supplementation in preventing vitamin D deficiency (NCT03536845). We included children with epilepsy aged 2–16 years who were treated with ASMs from December 2017 to March 2021. Children with preexisting vitamin D metabolism problems, vitamin-D-dependent rickets, malabsorption syndromes, renal disease, and hepatic disease were excluded. The baseline demographic data, anthropometric measurements, seizure types, epilepsy syndromes, ASMs, and seizure control measures were recorded. Blood tests for vitamin D (25-hydroxyvitamin D [25(OH)D), serum calcium, serum phosphorus, and parathyroid hormone levels were performed. Based on vitamin D concentration, patients were categorized as deficient (<50 nmol/L), insufficient (74.9–50 nmol/L), or normal (>75 nmol/L). Results: Of 159 recruited children, 108 (67.92%) had generalized seizures, 44 (27.67%) had focal seizures, and 7 (4.4%) had unknown onset seizures. The number of children receiving monotherapy was 128 (79.0%) and 31 (19.1%) children were receiving polytherapy. The mean vitamin D concentration was 60.24 ± 32.36 nmol/L; 72 patients (45.28%) had vitamin D deficiency and 45 (28.3%) had vitamin D insufficiency. No significant difference in vitamin D concentration was observed between children receiving monotherapy and those receiving polytherapy. The main risk factors of vitamin D deficiency were obesity and receiving enzyme-inducer ASMs. Conclusions: The prevalence of vitamin D deficiency was high among children with epilepsy. Obese children with epilepsy and those on enzyme-inducer ASMs were at increased risk for vitamin D deficiency. Further studies are needed to establish strategies to prevent vitamin D deficiency.

## 1. Introduction

Vitamin D deficiency is a global health concern, especially during rapid growth in infancy, childhood, and adolescence [1]. Beyond its well-known role in bone health, vitamin D is implicated in the modulation of physiological and pathological processes and the prevention and treatment of many immunological [2], cardiovascular [3], and neurological diseases, such as dementia [4], Parkinson’s disease [5], and multiple sclerosis [6]. Vitamin D deficiency is caused by the low levels of vitamin D in the diet, insufficient sunlight exposure, obesity, decreased mobility, geographic latitude, and genetic predisposition [7,8].

Epilepsy is one of the most common neurological disorders that affects people of all ages in about 0.5–1% of the population [9]. Patients with epilepsy have a significantly increased risk of developing bone metabolism abnormalities, which may cause rickets and osteomalacia, resulting in a two- to six-times higher risk of fracture than the general population and an increased incidence of osteoporosis in adult life [10]. In multiple cohort studies, antiseizure medications (ASMs) significantly lowered vitamin D concentration; this effect is attributed to the duration, the pharmacokinetics, and the number of ASMs [11,12,13].

The prevalence of vitamin D deficiency is highest in Asia, the Middle East, and Africa [14]. For instance, in Europe, 4–7% of school-age children suffer from vitamin D deficiency; however, in Saudi Arabia, it affects 45.5% of children aged between 6 and 15 years [15,16].

Although the association between vitamin D deficiency in patients with epilepsy is documented, routine screening for vitamin D deficiency and prophylactic supplementation of vitamin D in children with epilepsy is not evidence based. Furthermore, it is highly variable across clinical practice guidelines [17,18,19]. The randomized control trial of vitamin D supplementation for children with epilepsy was a novel six-month intervention that will provide evidence for vitamin D supplementation [20]. This paper presents the baseline characteristics of vitamin D supplementation for children with epilepsy in a randomized control trial. The cohort assessed the prevalence and the predictors of vitamin D deficiency in children with epilepsy aged 2 to 16 years old.

## 2. Materials and Methods

### 2.1. Population, Sampling, and Recruitment

This study analyzed the baseline data of a pragmatic randomized controlled open-labeled trial [20]. We recruited children aged 2–16 years diagnosed with epilepsy and treated with ASMs attending the outpatient pediatric neurology clinic in King Saud University Medical City in Riyadh, Kingdom of Saudi Arabia, from December 2017 to March 2021. The trial registration number on May 25, 2018, at clinicaltrials.gov is NCT03536845.

We excluded children with preexisting vitamin D metabolism problems, vitamin-D-dependent rickets, malabsorption syndromes (such as celiac disease, inflammatory bowel disease), renal disease, hepatic disease, baseline hypercalcemia with total corrected calcium greater than 2.5 mg/dL, vitamin D concentration greater than 250 nmol/L, and urine calcium to creatinine ratio greater than 1.2 mol/mol or 0.41 g/g.

During the trial screening, demographic data, medical history, and blood samples were taken from eligible, consenting participants. This paper analyzes and reports data collected at the screening visit from the pre-randomization phase.

### 2.2. Data Collection and Laboratory Procedures

Recruitment to the trial was completed in March 2021. Part of the trial recruitment was conducted during the COVID-19 lockdown period. We collected baseline demographic data and anthropometric measurements (height, weight, and body mass index (BMI)). Normal BMI was defined as being on the 5th to less than the 85th percentile [21]. Moreover, we reviewed the type of epilepsy, seizure control over the last six months, and ASMs. Children receiving two or more ASMs were considered to be receiving polytherapy. We performed the blood test for parathyroid hormone (PTH) and vitamin D (25-hydroxyvitamin D [25(OH)D) using electrochemiluminescence, serum calcium, and serum phosphorus. Based on the baseline concentration of vitamin D, patients were categorized as deficient (<50 nmol/L), insufficient (74.9–50 nmol/L), or normal (75 nmol/L) [22].

### 2.3. Ethical Consideration

Ethical approval for this study was taken from King Saud University Institutional Review Board. Furthermore, the study procedures complied with the Good Clinical Practice and Declaration of Helsinki. After explaining the study, we obtained consent and assent from the family during their initial clinic visit.

### 2.4. Statistical Analysis

Normally distributed continuous data were presented as mean with SD while the median (interquartile range) was used for non-parametric data. Dichotomous data were presented as frequency and percentage. We compared the groups using the Chi-square test for the impact of ASM type, monotherapy versus polytherapy, and BMI on vitamin D concentration. In contrast, we used the student *t*-test in the normally distributed continuous outcomes for independent samples and the Wilcoxon sign test for non-parametric data. All analysis was conducted using STATA SE version 16.

## 3. Results

In total, 159 children were recruited, with a mean age of 9.06 ± 3.35 years, of which 85 (53.4%) were males. Most parents had received a university education. The seizure types were generalized (108, 67.92%), unknown onset (7, 4.4%), and focal (44, 27.04%). Children receiving monotherapy were 128 (79.0%), while 31 (19.1%) were taking polytherapy (Table 1). Further, 104 (65.41%) children were seizure free for the previous six months.

The mean vitamin D concentration was 60.24 ± 32.36 nmol/L. Overall, 45 (28.3%) children had vitamin D insufficiency and 72 (45.28%) had vitamin D deficiency (Table 2). There was no significant difference in vitamin D concentration or the number of children affected by vitamin D insufficiency or deficiency between the monotherapy and polytherapy groups. Among the 71 (44.65%) who received previous treatment with vitamin D, 43 (27.04%) had a known family history of vitamin D deficiency. The mean reported sun exposure time was 0.558 h/day (SD ± 0.910). The mean time spent on physical activities was 1.21 h/day (SD ± 1.57). The main risk factors of vitamin D deficiency were obesity (odd ratio 2.29, 95% confidence interval 0.96–5.94, *p* = 0.04) and receiving treatment with enzyme-inducer ASM (OR 2.29, 95% CI 0.95–5.94, *p* = 0.04) (Table 3).

PTH mean concentration was 4.98 ± 3.21 pmol/L. The mean serum calcium was 2.34 ± 0.107 mmol/L and that of the phosphorus was 1.501 ± 0.225 mmol/L. The vitamin D concentration was not significantly correlated with phosphorus or calcium level. However, the correlation between vitamin D concentration and PTH was r^2^ = −0.34, *p* < 0.001. All patients did not have physical signs of rickets.

## 4. Discussion

Vitamin D deficiency remains a global health concern. In our cohort, we found that 73.6% of children have hypovitaminosis D, of whom 45.3% have vitamin D concentration < 50 nmol/L. This is the first study to assess the prevalence of vitamin D deficiency in children with epilepsy in Saudi Arabia. This high prevalence of hypovitaminosis D is alarming, given that 44% of our cohort have previously been treated for vitamin D deficiency and those children come from educated families living in urban areas.

Our cohort’s vitamin D deficiency status is relatively higher than what has been reported in other studies [12,23,24,25]. The prevalence of vitamin D deficiency among adults with epilepsy was 67.3% in a previous study [26]. Choong Yi Fong et al. found vitamin D deficiency in 22.5% of children with epilepsy and 19.7% had vitamin D insufficiency [12]. The overall high vitamin D deficiency can explain the high prevalence of vitamin D deficiency in our cohort in Saudi Arabia on top of the increased vitamin D requirement in children with epilepsy.

Although Saudi Arabia is near the equator line and has the luxury of sunshine, vitamin D deficiency is highly prevalent because of ethnic and environmental factors. First, there is a lack of naturally occurring food rich in vitamin D and a lack of mandatory food fortification policy with vitamin D [27]. Moreover, limited exposure to direct sunlight because of the high temperature reduces the exposed body areas with extensive clothing and the lower efficiency of vitamin D natural synthesis among people with darkly pigmented skin [18].

Similar to other studies, we found a higher rate of vitamin D deficiency among children receiving enzyme-inducer ASM and children with obesity. In a systematic review with meta-analysis, the prevalence of vitamin D deficiency among children with epilepsy treated with enzyme-inducer ASMs was found to be 33%, compared to 24% for those on non-enzyme inducers [28]. Another study reported the association between the use of enzyme-inducer ASMs and low vitamin D level in epilepsy patients [29]. Carbamazepine is one of the enzyme-inducer ASMs associated with lower levels of 25-hydroxyvitamin D in persons with epilepsy [30]. Enzyme-inducer ASMs work through CYP450 and potentially increase the hepatic metabolism of vitamin D, resulting in low vitamin D plasma levels [31]. Patients on enzyme-inducer ASMs need a higher vitamin D supplement dose than those receiving non-enzyme inducers [32].

Although the number of ASMs used by the child negatively influences the level of vitamin D [23], in our study, children on monotherapy or polytherapy had a similar risk for vitamin D deficiency. Similarly, many risk factors contribute to vitamin D deficiency. In our study, we did not find a significant association between vitamin D deficiency and gender, sun exposure, family history of vitamin D deficiency, previous treatment for vitamin D deficiency, or presence of developmental delay in the child. This is because of the lower overall sun exposure in the cohort and high vitamin D deficiency in the children and their families, possibly resulting in fallacy bias.

The limitations of our study include that we did not assess the impact of seasonal variation on the measured vitamin D concentration. We also did not evaluate the dietary intake of vitamin D and its effect on our patients’ vitamin D concentration. Although all patients did not have physical signs of rickets, we did not perform bone mineral density to assess the impact of vitamin D deficiency on bone mineralization. Moreover, the report of our cohort covered a single tertiary center. Therefore, further studies need to confirm our findings.

Vitamin D deficiency is common among Saudi children with epilepsy. Our findings suggest that clinicians should consider screening for vitamin D deficiency among children who are obese and use enzyme-inducer ASMs. Further studies are needed to establish strategies to prevent vitamin D deficiency.

## Figures and Tables

**Table 1 children-09-01696-t001:** Baseline characteristics of the participants.

Characteristics	Overall *n* = 159	Monotherapy *n* = 128	Polytherapy *n* = 31	*p*-Value
Mean age in years	9.06 ± 3.35	9.21 ± 3.16	8.48 ± 4.07	0.28
Gender, male	85 (53.46)	71 (55.47)	14 (45.16)	0.30
Preterm Pregnancy	16 (10.06)	11 (8.73)	5 (16.13)	0.22
Developmental delay	60 (37.73)	42(32.81)	18(58.06)	0.009
Obesity	54 (33.96)	45 (35.16)	9 (29.03)	0.52
Seizure type				0.05
Generalized	108 (67.92)	93 (72.66)	15 (48.38)	
Unknown onset	7 (4.40)	5 (3.91)	2 (6.67)	
Focal	44 (27.67)	30 (23.44)	14 (45.16)	
Seizure frequency				
Seizure-free	104 (65.41)	93 (72.66)	11 (35.48)	<0.001
Per day	8 (5.03)	5	3	0.003
Per week	14 (8.81)	8	6	0.344
Per month	31 (19.5)	21	10	0.720
Per 6 months	12 (7.55)	6	6	0.362
New ASM added	11 (6.92)	4 (3.17)	7 (22.58%)	<0.001
Living in urban area	144 (90.57)	115	29	0.57
Current dose increase	40 (25.16)	28 (22.22%)	12 (38.71%)	0.06
Family history of vitamin D deficiency	43 (27.04)	36 (28.12)	7 (22.58)	0.53
Previous treatment with vitamin D	71 (44.65)	60 (46.88)	11 (35.48)	0.25
Parent reported sun exposure, hr/day *	0.56± 0.91	0.601 ± 0.963	0.38 ± 0.62	0.24
Outdoor mean (SD) time hr/day *	3.11 ± 2.8	3.1 ± 2.84	3.12 ± 2.77	0.98
Screen time, hr/day *	2.91 ± 2.69	2.95 ± 2.72	2.70 ± 2.57	0.64
Physical activity time, hr/day *	1.21 ± 1.57	1.13 ± 1.44	1.56 ± 2.00	0.18
Maternal education **				0.27
Elementary	10 (6.29)	6 (4.76)	4 (12.90)	
Intermediate	17 (10.69)	15 (11.90)	2 (6.45)	
Secondary	35 (22.01)	28 (22.22)	7 (22.58)	
Diploma	8 (5.03)	5 (3.97)	3 (9.68)	
University	81 (50.94)	66 (52.38)	15 (48.39)	
Illiterate	6 (3.77)	6 (4.76)	0 (0)	
Paternal education **				0.99
Elementary	6 (3.77)	5 (3.97)	1 (3.23)	
Intermediate	17 (10.69)	14 (11.11)	3 (9.68)	
Secondary	37 (23.27)	29 (23.02)	8 (25.81)	
diploma	10 (6.29)	8 (6.35)	2 (6.45)	
University	85 (53.46)	68 (53.97)	17 (54.84)	
Illiterate	2 (1.26)	2 (1.59)	0 (0)	
Total Calcium *	2.34 ± 0.107	2.34 ± 0.111	2.31 ± 0.086	0.15
Phosphorus *	1.501 ± 0.225	1.504 ± 0.209	1.49 ± 2.866	0.79
PTH *	4.98 ± 3.21	4.946 ± 2.844	5.127 ± 4.491	0.79
Urine calcium: creatinine ratio	0.34 ± 0.65	0.35 ± 0.71	0.27 ± 0.27	0.62

Values are *n* (%) unless indicated; * mean ± SD; ** 2 patients with missing parental education.

**Table 2 children-09-01696-t002:** Vitamin D level according to antiseizure medication type.

	Overall *n* = 159	Monotherapy *n* = 128	Polytherapy *n* = 31	*p*-Value
Vitamin D level, nmol/L *	60.24 ± 32.36	58.31 ± 28.23	68.18 ± 45.36	0.1276
Vitamin D > 75 nmol/L	42 (26.4)	32 (25)	10 (32.26)	0.35
<75 nmol/L	117 (73.6)	96 (75)	21 (67.74)	0.411
<50 nmol/L	72 (45.28)	59 (46.09)	13 (41.94)	0.676

Values are *n* (%) unless indicated; * mean ± SD.

**Table 3 children-09-01696-t003:** Distribution of risk factors predisposing for Vitamin D deficiency.

	Overall *n* = 159	Vitamin D > 75 nmol/L *n* = 42	Vitamin D < 75 nmol/L *n* = 117	*p*-Value
Family history of vitamin D deficiency	43 (27.04)	9 (21.42)	34 (29.05)	0.340
Previous treatment with Vitamin D	71 (44.65)	21 (50)	50 (42.73)	0.41
Developmental delay	60 (37.73)	18 (42.85)	42 (35.89)	0.42
Obesity	54 (33.96)	9 (21.42)	45 (38.46)	0.046
Enzyme-inducer	54 (33.96)	9 (21.42)	45 (38.46)	0.046
Parent reported sun exposure, hr/day *	0.56 ± 0.91	0.7 ± 1.4	0.6 ± 0.8	0.42
Outdoor time hr/day *	3.11 ± 2.8	3.3 ± 2.7	3.05 ± 2.8	0.66

Values are *n* (%); * mean ± SD.

## Data Availability

Not applicable.

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
