# Peer review of "Prevalence and Related Risk Factors of Vitamin D Deficiency in Saudi Children with Epilepsy"

_children, 2022, doi:10.3390/children9111696_

Round 1

Reviewer 1 Report

Comments inserted in pdf file. Please see and consider additional analyses.

Author Response

Dear editor, Many thanks for the feedback regarding the submitted manuscript. I would like first to thank the Reviewers for the effort and the pertinently raised questions. Below, are the answers to the points raised:

has a role

Response: Changed

Kindly use ILAE 2017 classification for seizures.

Response:  Changed

What was the duration of disease/epilepsy?

Response:  We did not collect information on the disease duration.

Were these newly diagnosed patients or prevalent cases?

Response:  Those are prevalent cases

Why didn't we exclude CP?

Response:  It is known to be associated with low vitamin D and calcium levels.

How were these excluded? You mean to say 'suspected' malabsorption or 'symptomatic' malabsorption?

Response:  We excluded children with preexisting malabsorption.

What if epilepsy had been diagnosed <6 months back?

Response:  Those children were included in the study.

What about testing for renal disease, hepatic disease, and urine 78 calcium to creatinine ratio - which were exclusion criteria?

Response: At baseline, we performed renal function, liver function, and urine calcium to creatinine ratio, if those were abnormal, the patient was excluded.

Total sunlight exposure/outdoor play, an important determinant of vitamin D levels in tropical countries, could have been assessed.

Response: We added those risk factors in table 3.

Is analysis for individual drugs available?

Response: It is available in detail, but we prefer not to add it. The following medications were used: Valproic acid, Lamotrigine, Levetiracetam, Phenobarbitone, Topiramate, Ethusuxamide, Clonazepam, Clobazam, and Vigabatrin

The figure does not have any relevance to the study objectives or baseline data.  Response:  The figure removed

Thank you again for your feedback.

Sincerely, Fahad

Reviewer 2 Report

Few comments are listed here for the authors’ consideration to further improve the quality and overall impact of the manuscript.

Authors should include at the title that this study was in children with idiopathic epilepsy, an also in a population of Saudi children, with the purpose of emphasizing that it is the first study that shows this vitamin D deficiency in these epileptic children.

Which ASMs were used as treatment for idiopathic epilepsy in these children?

Which of these ASMs were found to be enzyme inducers or non-enzyme inducers? And which drugs were used as monotherapy or polytherapy in children? Showing a table is recommended.

Which new ASMs were added? table 1.

What was the previous treatment with vitamin D? indicated in table 1, and how long was it administered in these children?

What is the physiological meaning of the correlation between vitamin D deficiency and parathyroid hormone?

The frequency of seizures did not have any correlation with the vitamin D deficiency in this study?

Was vitamin D status was defined according to some Endocrine Guidelines? Or authors defined the deficiency (< 50 nmol/l), insufficiency (74.9-50 nmol/l) or normality (75 nmol/l) of vitamin D levels in their study?

At table 3, authors give “Distribution of risk factors predisposing for Vitamin D deficiency”, but why not include information of the group with deficiency (<50 nmol/l)?

Were the levels of vitamin D determined daily during the complete study? or when was the analysis performed?

There are two current references that authors could include in their study:

Cunha IA, Saraiva AM, Lopes P, Jesus-Ribeiro J, Duarte C, Leitão F, Sales F, Santana I, Bento C. Vitamin D deficiency in a Portuguese epilepsy cohort: who is at risk and how to treat. Epileptic Disord. 2021 Apr 1;23(2):291-298.

Likasitthananon N, Nabangchang C, Simasathien T, Vichutavate S, Phatarakijnirund V, Suwanpakdee P. Hypovitaminosis D and risk factors in pediatric epilepsy children. BMC Pediatr. 2021 Oct 2;21(1):432.

Author Response

Dear editor, Many thanks for the feedback regarding the submitted manuscript. I would like first to thank the Reviewers for the effort and the pertinently raised questions. Below, are the answers to the points raised:

Few comments are listed here for the authors’ consideration to further improve the quality and overall impact of the manuscript.

Authors should include at the title that this study was in children with idiopathic epilepsy, an also in a population of Saudi children, with the purpose of emphasizing that it is the first study that shows this vitamin D deficiency in these epileptic children.

Response: done

Which ASMs were used as a treatment for idiopathic epilepsy in these children?

Response: The following medications were used: Valproic acid, lamotrigine, levetiracetam, phenobarbitone, topiramate, ethosuximide, clonazepam, clobazam, and vigabatrin.

Which of these ASMs were found to be enzyme inducers or non-enzyme inducers? And which drugs were used as monotherapy or polytherapy in children? Showing a table is recommended.

Which new ASMs were added? table 1.

Response:  newly added ASMs were according to the child-specific treatment needs, those were prescribed by the primary neurologist. 

What was the previous treatment with vitamin D? indicated in table 1, and how long was it administered in these children?

Response:  We did not collect data on the type of vitamin D treatment pre-enrollment to the trial, nor have access to the length of treatment before.

What is the physiological meaning of the correlation between vitamin D deficiency and parathyroid hormone?

Response:  previous literature has elucidated the fact that low vitamin D concentration leads to increased PTH levels leading to negative bone health, and hence some of the world experts have recommended some of the cutoffs used nowadays to define deficiency and treatment threshold based on PTH suppression.

The frequency of seizures did not have any correlation with vitamin D deficiency in this study?

Response:  Yes, vitamin D concentration was not associated with the frequency of seizures from the baseline data. However, this association will need to be further confirmed in the intervention.

Was vitamin D status defined according to some Endocrine Guidelines? Or authors defined the deficiency (< 50 nmol/l), insufficiency (74.9-50 nmol/l), or normality (75 nmol/l) of vitamin D levels in their study?

Response:  We defined those cutoffs according to the Institute of Medicine (IOM). We added the reference to the methods section

At table 3, authors give “Distribution of risk factors predisposing for Vitamin D deficiency”, but why not include information of the group with deficiency (<50 nmol/l)?

Response:  the group that is <75nmol/l includes the data of those who have vitamin D concentration <50nmol/l.

Were the levels of vitamin D determined daily during the complete study? or when was the analysis performed?

Response:  Those vitamin D levels were done upon recruitment to the study at baseline before randomization in the trial.

There are two current references that authors could include in their study:

Cunha IA, Saraiva AM, Lopes P, Jesus-Ribeiro J, Duarte C, Leitão F, Sales F, Santana I, Bento C. Vitamin D deficiency in a Portuguese epilepsy cohort: who is at risk and how to treat. Epileptic Disord. 2021 Apr 1;23(2):291-298.

Likasitthananon N, Nabangchang C, Simasathien T, Vichutavate S, Phatarakijnirund V, Suwanpakdee P. Hypovitaminosis D and risk factors in pediatric epilepsy children. BMC Pediatr. 2021 Oct 2;21(1):432.

 Response:  Added to the revised version of the manuscript

Thank you again for your feedback.

Sincerely, Fahad

Round 2

Reviewer 1 Report

None.